# Speed dependent descending control of freezing behavior in *Drosophila melanogaster*

Ricardo Zacarias[1], Shigehiro Namiki[2], Gwyneth M. Card[2], Maria Luisa Vasconcelos[1] & Marta A. Moita[1]

The most fundamental choice an animal has to make when it detects a threat is whether to freeze, reducing its chances of being noticed, or to flee to safety. Here we show that *Drosophila melanogaster* exposed to looming stimuli in a confined arena either freeze or flee. The probability of freezing versus fleeing is modulated by the fly's walking speed at the time of threat, demonstrating that freeze/flee decisions depend on behavioral state. We describe a pair of descending neurons crucially implicated in freezing. Genetic silencing of DNp09 descending neurons disrupts freezing yet does not prevent fleeing. Optogenetic activation of both DNp09 neurons induces running and freezing in a state-dependent manner. Our findings establish walking speed as a key factor in defensive response choices and reveal a pair of descending neurons as a critical component in the circuitry mediating selection and execution of freezing or fleeing behaviors.

[1] Champalimaud Research, Champalimaud Centre for the Unknown, 1400-038 Lisbon, Portugal. [2] Janelia Research Campus, Howard Hughes Medical Institute, Ashburn, VA 20147, USA. These authors contributed equally: Maria Luisa Vasconcelos, Marta A. Moita. Correspondence and requests for materials should be addressed to M.A.M. (email: marta.moita@neuro.fchampalimaud.org)

Animals rely on similar kinds of cues, such as visual looming stimuli, to detect a predator's rapid approach[1–3]. Although great progress has been made in the study of threat detection mechanisms, much less is known regarding when and how different defensive behaviors are performed. Animals employ several defensive strategies, grossly categorized into flight, freeze and fight. The expression of these behaviors is modulated by external factors such as the presence of offspring, access to shelter and the animal's internal state[4–11]. How each specific defensive response is selected and executed remains unclear.

In mammals, mostly from studies using rodents, multiple brain regions have been implicated in the expression of freezing and escape responses, including the amygdala, hypothalamus and peri-aqueductal grey[12–17]. More recently, microcircuits within these brain regions that regulate the expression of flight or freeze behaviors have been characterized[18,19]. Much less is known about the mechanisms of expression of defensive behaviors in other vertebrates. In fish, Mauthner cells, a pair of large neurons in the hindbrain, have been implicated in fast escape responses (reviewed in ref. [20]), whereas other spinal cord projecting neurons are involved in slower escapes[21]. Even though the zebrafish habenula has been shown to down regulate freezing, favoring escape responses, how freezing behavior is produced remains unknown. In invertebrates, research has focused more on the mechanisms of escape responses. Fruit flies can exhibit either fast or slow looming-triggered takeoff responses, corresponding to different modules that can be used flexibly with the former relying on the giant fiber and the latter on other descending neurons[1,22]. Although freezing has been reported in fruit flies exposed to an inescapable visual threat[23], a systematic quantification of this behavior and exploration of its neural underpinnings is lacking. Where the focus of research concerning defensive behaviors in mammals has centered around learned freezing responses, innate escape responses have received more attention in other organisms. In addition, how external or internal factors impinge on these circuits to regulate choice between different responses remains largely unknown. To address this issue, we decided to use *Drosophila melanogaster* for its

arsenal of genetic tools to dissect neural circuits and the ability to use large sample sizes allowing for detailed quantitation of behavior. We developed a visual assay to track the responses of flies to an expanding shadow that mimics a large object on a collision course. This stimulus, known as looming, triggers defensive behaviors in virtually all visual animals tested, including fruit flies[1–3]. Flies were exposed to multiple looming stimuli in an enclosed arena to increase the likelihood of seeing both escape and freezing responses. In our experimental set-up, sustained freezing is the predominant defensive response. Flies that do not freeze display escape responses directed away from the looming stimulus. Taking advantage of a closed-loop system, which allows for the presentation of visual stimuli dependent on the behavior of flies, we found that the decision to freeze or flee is modulated by movement speed at the time of threat. Through genetic manipulation of neuronal activity we identified a pair of descending neurons whose activity is required for freezing. Moreover, their ability to drive freezing, through optogenetic activation, depends on the movement speed of flies at the time of stimulation. These results reveal that innate responses to threats can be modulated by behavioral state and identifies an element of the freezing circuit that is susceptible to this modulation.

## Results

**Flies rarely jumped to repeated inescapable looming.** Figure 1a shows a schematic representation of the experimental setup. We placed single flies in a covered walking arena and gave them 5 min to explore. A computer monitor angled above the arena showed a looming stimulus (black circle expanding on a white background) repeated 20 times over a subsequent 5 min period. As a control, we showed a separate group of flies a sequence of randomly appearing black dots resulting in a similar change in luminance but with no pattern of expansion (Fig. 1b). Notably, flies could not escape from the arenas.

We found that looming stimuli only occasionally triggered escape jumps (6.4% of looming stimuli, 384/6000). These jumps likely correspond to takeoff attempts, the most studied defensive response in insects[22,24–32]. The number of jumps per fly was

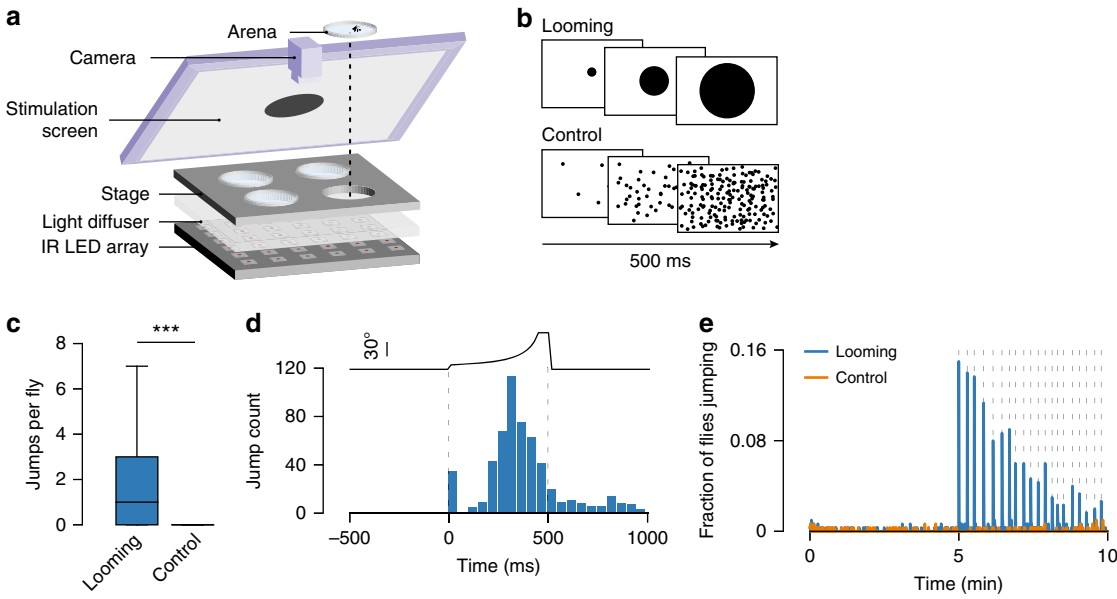

**Fig. 1** - Flies jump in response to repeated looming. **a** Schematic of behavioral assay. **b** Schematic of visual stimuli, 300 flies were tested in each condition. **c** Number of jumps detected per fly during the 5 min stimulation period. Center line, median; box limits, upper (75) and lower (25) quartiles; whiskers, 1.5x interquartile range. **d** Jump timing within a 1 s window around looming stimuli. Dashed lines indicate looming onset and offset. Top, size of looming disk (visual angle). **e** Proportion of flies jumping throughout the 10 min session. Dashed lines indicate stimulus presentations. *** denotes $p < 0.001$

significantly higher for the looming (median = 1, interquartile range (IQR) = 0–3) than the control (median = 0, IQR = 0) condition (Wilcoxon rank-sum test, $p < 0.001$, Fig. 1c). The large majority of these events occurred within the window of stimulation, before the circle reached its maximum size (Fig. 1d). The efficacy of looming stimuli to elicit jumps was lower in our experimental conditions than those previously reported[27,30], where flies were exposed to a single escapable looming. To confirm that with our experimental setup and stimulus, we can elicit a high rate of looming-triggered takeoff, we allowed flies to climb to the top of the arena through a hole in the lid and presented a looming. In this condition 90% (36/40) of flies jumped in response to the stimulus. Finally, we found that the probability of jumping to repeated looming decreased over the course of the 20 stimulus presentations (Fig. 1e), suggesting that with multiple presentations flies may have habituated to looming. Alternatively, flies could be adopting other defensive strategies.

**Most flies responded to looming with sustained freezing**. To determine whether flies displayed alternative responses to looming, we analyzed fly speed over the course of the experiment. The largest fraction of flies decreased their speed (Fig. 2a). Manual inspection of the videos led to the observation that flies were not just walking slower nor grooming, they were completely immobile, i.e., freezing. In many cases, these immobile flies sustained unusual postures for long periods of time, including postures with legs off the ground (Supplementary Movies 1 and 2). In order to quantify freezing, we created an automated classifier based on pixel change recorded in a region of interest surrounding the fly (Supplementary Fig. 1).

The fraction of flies freezing increased gradually with each looming presentation, arguing against habituation of looming stimulus detection (Fig. 2b). By the end of the stimulation period, 70% (210/300) of flies were freezing compared to 12% (36/300)

for the control stimulus ($X^2$ test, $p < 0.001$, Fig. 2b). Flies tested without any stimulation showed freezing levels similar to control (Supplementary Fig. 2). Interestingly, there was a sharp decrease in freezing during each stimulus presentation. Further examination showed that freezing flies displayed startle responses during looming, but quickly returned to an immobile state (Supplementary Movie 1). These responses were characterized by a spike in pixel motion that peaked at the end of the expansion of the looming stimulus (Fig. 2c). These startle responses may correspond to an aborted jump[31]. In addition, we observed a bimodal distribution of freezing, where most flies either froze for long periods of time (>3 min) or did not freeze at all, with fewer flies displaying intermediate freezing levels (Fig. 2d. Probability density function is described in the Methods section). Hence, flies froze for long periods of time only breaking freezing for the brief moments of startle. To test whether prolonged freezing was caused by repeated stimulation, we exposed flies to five looming stimuli over the course of one minute and asked whether flies would resume locomotion during the remaining 4 min of the session. We found that a substantial fraction of flies continued freezing even after the stimulation ended (Fig. 2e). Half (50/100) of the flies froze for longer than one minute and some flies sustained freezing up to 5 min (Fig. 2f). These long bouts of freezing are in sharp contrast with previous studies, which have only reported short-lived freezing bouts of up to a few seconds in *Drosophila*[23,31,33].

**Flies that did not freeze, fled instead**. Since running is an alternative form of defensive behavior[23,34], we next analyzed locomotor behavior excluding all freezing and grooming bouts, hence only periods classified as walking (>4 mm s$^{-1}$). Walking speed gradually decreased during the baseline period, reflecting a common process of habituation to the test arena[35]. However, during stimulation, walking speed increased relative to baseline

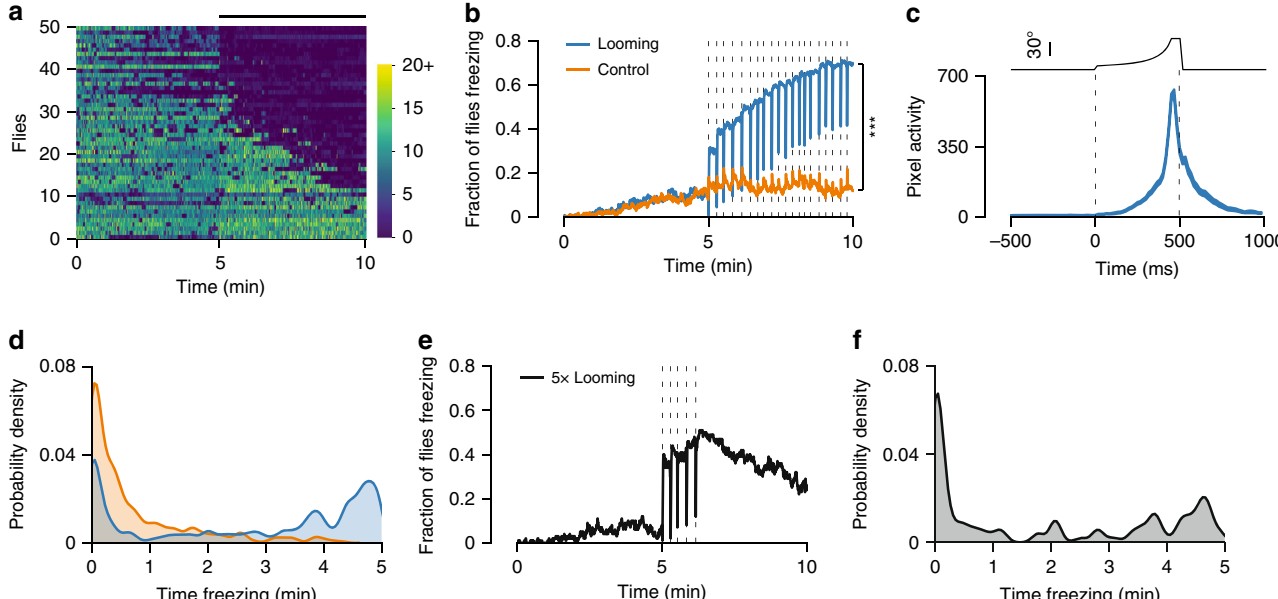

**Fig. 2** Flies freeze upon looming stimulation. **a** Speed raster for a random subset of 50 flies ordered by average speed during stimulation (ascending). Each row corresponds to one fly and each vertical line to 500 ms bins. Bar on top indicates the 5 min stimulation period. Color bar values refer to mm s$^{-1}$. **b** Proportion of freezing flies ($n = 300$ for each condition). Dashed lines represent stimulus presentations. **c** Startle behavior during freezing. Average (±s.e.m.) number of pixels changing around the fly in a 1 sec window around looming stimuli, including only trials where flies were freezing before and after the stimulus ($n = 1434$). Dashed lines indicate looming onset and offset. Top, size of looming disk (visual angle). **d** Distribution of individual flies by time spent freezing during the stimulation period. **e** Proportion of freezing flies to a shorter stimulation (5 looming presentations, $n = 100$). Dashed lines represent looming presentations. **f** Distribution of individual flies by time spent freezing during the stimulation period for the shorter stimulation experiment. *** denotes $p < 0.001$

(median = 1.47 mm s$^{-1}$, IQR = 0.67–2.71. One-sample Wilcoxon signed-rank test, $p < 0.001$). This effect was not observed in flies exposed to control stimuli, which further decreased their speed (median = −0.86 mm s$^{-1}$, IQR = −1.59 − (−0.26). One-sample Wilcoxon signed-rank test, $p < 0.001$, Fig. 3a, b). The average difference in walking speed between stimulation and baseline periods was significantly higher for flies exposed to looming relative to control (Wilcoxon rank-sum test, $p < 0.001$, Fig. 3b). In addition, we observed sharp increases in speed at the time of each looming presentation (Fig. 3a). We next asked whether the running bouts reported here correspond to escapes away from the threat. We measured the orientation of the paths of walking flies before and after each looming presentation (Fig. 3c). Before looming, paths in all orientations could be seen (median = 193.8°, IQR = 99.12–280.95, Fig. 3d). Upon looming, we observed a significant increase in orientation bias toward the side of the chamber furthest away from the source of the threat (median = 240.05°, IQR = 178.08–290.88, Wilcoxon signed-rank test,

$p < 0.001$, Fig. 3e). This suggests that flies heading towards the screen changed direction, while flies heading away from the screen increased their walking speed and maintained course. In contrast, the distribution of orientations did not change with presentation of the control stimulus (before control stimulus median = 182.39°, IQR = 91.82–274.13; after control stimulus median = 184.18°, IQR = 90–275.12. Wilcoxon signed-rank test, $p = 0.51$, Supplementary Fig. 3a). These findings indicate that the running bouts triggered by looming stimuli are not just a simple increase in locomotion but constitute directed escape responses.

To examine in more detail the temporal profile of these escape responses we plotted the average speed of walking flies aligned to looming onset (Fig. 3f, Supplementary Movie 3). We found that walking speed was relatively constant before the stimulus (Fig. 3f, marker #1). Upon looming onset, flies first sharply decreased their translational speed after which they showed a rapid burst of locomotion (Fig. 3f, marker #2 and 3). Walking speed remained elevated after the looming stimulus (Fig. 3f, marker #4). Next, we

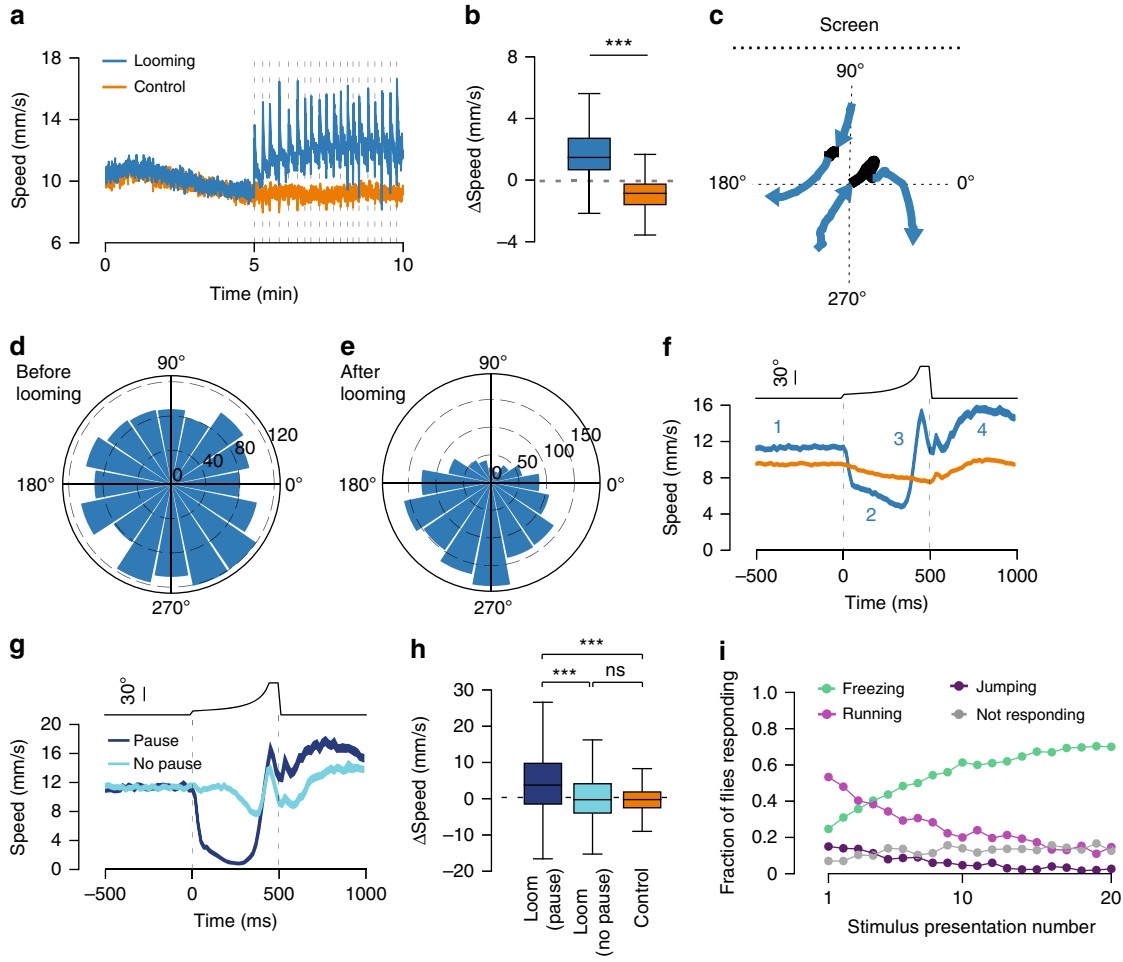

**Fig. 3** Flies flee to looming stimuli. **a** Average (±s.e.m.) fly speed including only time periods classified as walking. Dashed lines indicate stimulus presentations (in **a**, **b** looming $n = 280$ flies, control $n = 298$ flies). **b** Change in walking speed caused by stimulation (baseline period subtracted from stimulation period). **c** Example trajectories of walking trials. Black section corresponds to looming window and blue sections 500 ms before and after looming. **d**, **e** Distribution of path orientations before (d) and after (e) looming. Bar height indicates counts. Stimulus source (screen) was located at 90°. In **d**–**f**, only looming events where flies were walking before and after the stimulus were included (looming $n = 1574$ trials, control $n = 2881$ trials). **f** Looming-triggered speed profile. Average (±s.e.m.) speed in a 1 s window around looming for all walking trials. Blue numbers represent four stages of the response to looming: 1- pre-looming, 2- pause, 3- run, 4- post-looming. Top, size of looming disk (visual angle). **g** Looming-triggered speed profile of walking trials from the looming condition separated into responses that included a pause (dark blue, $n = 806$) and responses that did not (light blue, $n = 767$). **h** Change in walking speed caused by stimulus presentation (pre-looming period subtracted from post-looming period). **i** Fraction of flies performing the described behaviors for each of the 20 looming presentations. *** denotes, $p < 0.001$, ns not significant. Box plot elements: center line, median; box limits, upper (75) and lower (25) quartiles; whiskers, 1.5× interquartile range

examined speed profiles on a trial-by-trial basis and found that the decrease in speed observed (Fig. 3f, marker #2) was caused by a complete stop in some of the trials rather than a slowing down in all trials. Having defined a set of criteria to establish whether a fly paused (Supplementary Fig. 3b-d), we found that pauses were present in 51.2% (806/1573) of escape trials. Importantly, flies oriented away from the screen both in trials where they paused and trials they did not (Supplementary Fig. 3e, f) suggesting that pausing is not an obligatory element of an escape response. Next, we separated escape trials with and without pauses and examined the temporal speed profile of these two responses (Fig. 3g). We found that the increase in speed after looming relative to the speed before was significantly higher for flies exposed to looming that paused (median = 3.77 mm s$^{-1}$, IQR = −1.48–9.76) than flies that did not pause (median = −0.27 mm s$^{-1}$, IQR = −3.96–4.13) and flies exposed to the control stimulus (median = −0.25 mm s$^{-1}$, IQR = −2.5–1.86, Kruskal–Wallis and post-hoc Dunn tests, $p < 0.001$. No difference was found between flies that did not pause and control stimulus, $p = 0.27$, Fig. 3h). The pause in escape trials we observe may correspond to the brief immobility bouts before looming triggered take offs described previously[31].

Altogether, these results show that flies that did not freeze reliably escaped. Still there might be a fraction of flies that did not respond to the looming stimulus. To address this issue, we analyzed the fraction of flies freezing, jumping and running and estimated the fraction of flies that showed none of the above responses. Only a small fraction of flies did not respond to looming, showing a modest increase throughout the stimulation period. Furthermore, the decrease in escape responses is accompanied by a commensurate increase in the fraction of flies freezing (Fig. 3i). These results argue once more against a habituation process occurring during the repeated looming stimulation.

**Freezing/fleeing decisions were modulated by walking speed.** Our data suggest that in our experimental conditions flies select between two distinct behavioral strategies, freezing or fleeing. One possibility is that the time flies have to escape, which depends on the latency of threat detection, dictates whether an escape attempt or freezing is selected. In this scenario, when detection is slow there is less time left to escape thus freezing becomes more likely. It is possible to estimate the moment of looming detection in freezing trials as there is a sharp deceleration during the looming stimulus (Fig. 4a). For escape trials, the onset of the pause can be used to determine time of detection. We found no difference in the onset of deceleration in both freezing and escape trials (see Methods, randomization test, $p = 0.94$, Fig. 4a). However, this analysis revealed a striking difference in the average speed before looming. To further explore this observation, we sorted looming trials by the speed of flies 500 ms before looming onset and calculated the probability of freezing at different movement speeds. We observed a sharp decay in freezing probability with increasing speed, such that flies moving slowly or grooming were more likely to freeze upon looming stimulation than flies moving faster (Fig. 4b). To test this relationship, we designed a closed-loop experiment where the position of the tested fly was tracked online and looming stimuli were delivered at specific speed thresholds (see Methods). One group of flies received looming at low movement speeds and another at high movement speeds (Supplementary Fig. 4). Importantly, there was no difference in the average baseline speed between the two groups, indicating that their overall state was similar (high speed = 10.96 ± 0.53 mm s$^{-1}$, low speed = 10.92 ± 0.47 mm s$^{-1}$. Student's $T$ test, $p = 0.95$, Fig. 4c). The fraction of flies freezing was higher for flies exposed to looming stimuli when moving at lower speed (76.7% (46/60) vs. 26.8% (15/56) for high speed, $X^2$ test, $p < 0.001$, Fig. 4d). In addition, flies in the low speed group froze more (median = 72.94, IQR = 47.15–86.67) than flies in the high speed group

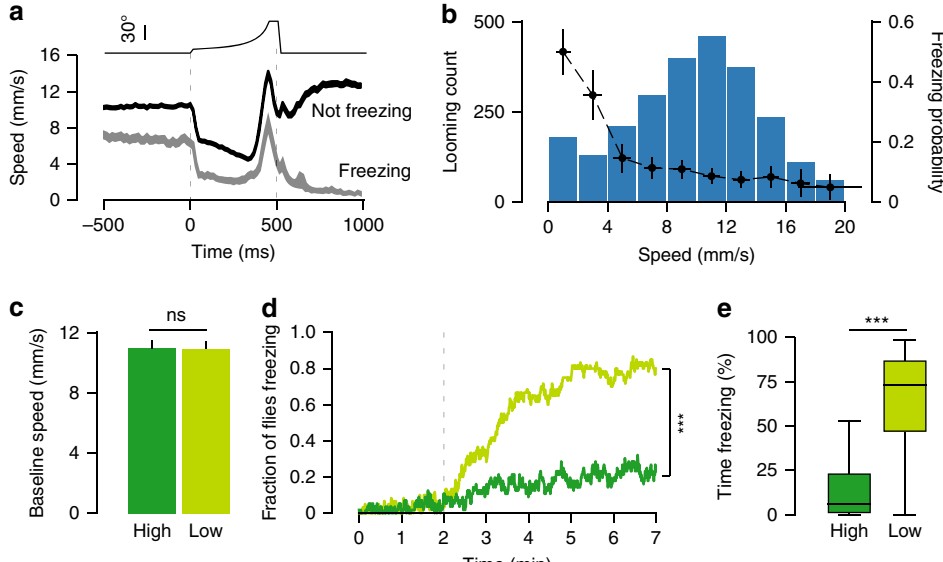

**Fig. 4** Response to looming is modulated by walking speed. **a** Looming-triggered speed profile (average ± s.e.m.) including only transition trials where flies were not freezing before the stimulus ($n = 2501$ loomings), separated into events that led to freezing (light gray, $n = 346$) and events that did not (dark gray, $n = 2155$). **b** Distribution of fly speed 500 ms before loomings where flies were not freezing (blue bars, $n = 2501$ loomings) overlaid with the probability of freezing after the stimulus for each speed interval (black dots and trace). X-error bars shows standard deviation (s.d.) for speed sampled within each interval and Y-error bars show 95% confidence intervals (CI). **a**, **b** Show data from the experiment in Figs. 1–3, whereas panels **c–e** show data from closed-loop experiments. In **c–e**, dark green corresponds to high speed ($n = 56$) group and light green to low speed group ($n = 60$). **c** Average + s.e.m. speed during baseline period for high and low speed groups. **d** Proportion of freezing flies. Dashed line indicates onset of stimulation. **e** Percent time spent freezing during stimulation period. *** denotes $p < 0.001$, ns not significant. Box plot elements: center line, median; box limits, upper (75) and lower (25) quartiles; whiskers, 1.5× interquartile range

(median = 5.99, IQR = 1.23–22.9, Wilcoxon rank-sum test, $p < 0.001$, Fig. 4e), thus confirming a modulation of freezing probability by the flies' movement speed. A link between locomotor activity and defensive behaviors has been previously reported in a study showing that low activity voles exposed to owl calls froze whereas high activity voles fled[36]. Inertia is one possible explanation for this effect, such that the faster an animal walks the more difficult is to come to a halt, explaining the sharp decrease of freezing with increased speeds. However, when we examined fleeing trials we found that the probability of pausing upon looming was not significantly modulated by walking speed ($r^2 = 0.17$, $p = 0.35$, Supplementary Fig. 4). Hence, the modulation of freezing probability is unlikely to result from a simple inability to become immobile at higher walking speed. Finally, the differential modulation of freezing and pausing by movement speed indicates that looming-triggered pausing and sustained freezing are mediated by different mechanisms.

**Activity of DNp09 is required for freezing but not fleeing.** Next, we searched for the neural mechanisms underlying the

defensive behaviors observed. We focused on freezing, as it corresponds to the dominant behavior adopted by flies in our experimental conditions, is readily quantifiable and, although it is conserved across the animal kingdom[36–38], very little is known regarding the neural mechanisms of freezing in insects. We performed an unbiased screen, testing for looming triggered freezing of fly lines expressing a hyperpolarizing potassium channel, Kir2.1[39], in different subsets of descending neurons (DNs)[40]. We focused on DNs, as these convey information from the brain to the ventral nerve cord, being therefore good candidates for the control of behavior. From this screen we identified a bilateral pair of DNs, DNp09, with dendrites innervating the posterior protocerebrum and a large axon extending throughout the ventral nerve cord, as well as the posterior slope and gnathal ganglion in the brain (Fig. 5a. Weak off-target labeling can be verified on the FlyLight project website: http://splitgal4.janelia.org/cgi-bin/splitgal4.cgi, line SS01540). Silencing these neurons significantly decreased the occurrence (DNp09 > Kir2.1: 20% (12/60); DNp09/ + : 55% (33/60); Kir2.1/ + : 63.3% (38/60), $X^2$ test, $p < 0.001$. No difference was found among the parental controls, $p = 0.46$, Fig. 5b) and duration of freezing relative to the

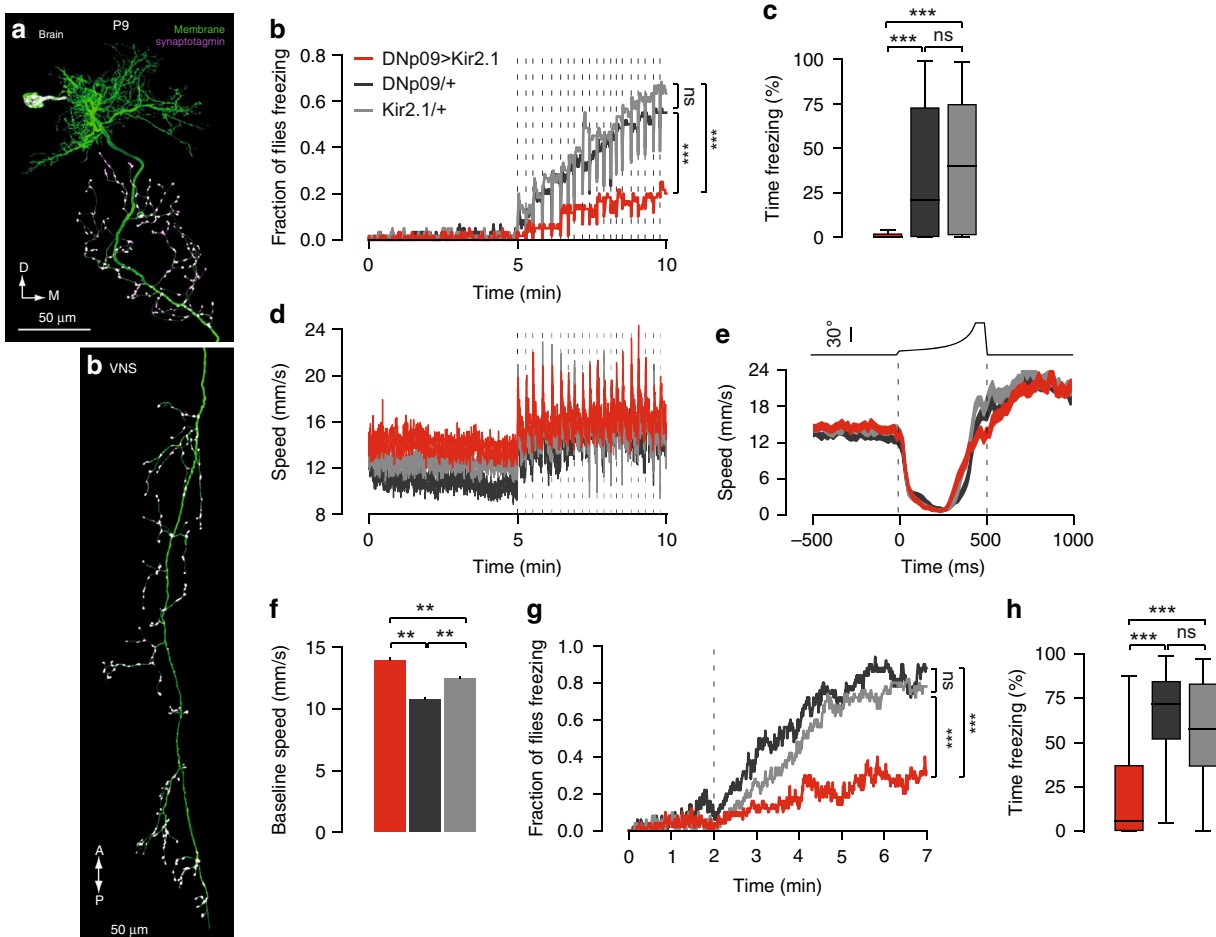

**Fig. 5** Silencing DNp09 neurons disrupts freezing but not running. **a** DNp09 morphology. DNp09 split-GAL4 driving membrane-bound GFP (green) combined with anti-synaptotagmin-HA staining (purple). In **b–h**, red corresponds to DNp09 > Kir2.1 flies, dark gray to DNp09/ + flies and light gray to Kir2.1/ + flies. In **b–d**, $n = 60$ flies for each condition. **b** Proportion of freezing flies. Dashed lines represent stimulus presentations. **c** Percent time spent freezing during stimulation period. **d** Average (±s.e.m.) fly speed including only time periods classified as walking. Dashed lines indicate stimulus presentations. **e** Looming-triggered speed profile (average ± s.e.m.) of walking trials that include a pause (DNp09 > Kir2.1 $n = 234$, DNp09/ + $n = 253$, Kir2.1/ + $n = 259$). **f** Average ( + s.e.m.) speed during baseline period ($n = 60$ for all conditions). In **g–h**, $n = 50$ flies for each group. **g** Proportion of freezing flies for low speed closed-loop looming stimulation. Dashed line indicates onset of stimulus presentations. **h** Percent time spent freezing during stimulation period for closed-loop experiment. ** denotes $p < 0.01$, *** denotes $p < 0.001$, ns not significant. Box plot elements: center line, median; box limits, upper (75) and lower (25) quartiles; whiskers, 1.5x interquartile range

parental controls (DNp09 > Kir2.1: median = 0.17, IQR = 0–1.67; DNp09/ + median = 20.58, IQR = 0.33–76.3; Kir2.1/ + median = 39.75, IQR = 1.38–74.5. Kruskal–Wallis and post-hoc Dunn tests revealed a significant difference between DNp09 > Kir2.1 and both parental controls, $p < 0.001$. No difference was found among the parental controls, $p = 1$, Fig. 5c). This effect was not due to an overall decrease in sensitivity to looming since running was intact (Fig. 5d and Supplementary Fig. 5a and b). Furthermore, the frequency of jumps was increased, raising the possibility that in control flies freezing behavior directly or indirectly inhibits jumping (Supplementary Fig. 5c). These results were replicated in a different genetic background (Supplementary Fig. 6).

Although looming-triggered pausing and freezing seem to be mediated by different mechanisms, these might be partially overlapping such that activity of DNp09 might contribute to both. DNp09-silenced flies that ran in response to looming still exhibited a pause upon looming onset (Fig. 5e), albeit less frequently (DNp09 silenced paused in 28.5% (234/822) of the trials compared to 46.5% (259/816) and 44.0% (253/573) in parental controls). This effect on pausing frequency was less robust than the effect on freezing. For example, when testing wild-type Dickinson Lab (DL) flies as parental controls, the effect of silencing DNp09 on pausing frequency was not reliable as it did not differ from one of the parental controls (DNp09-silenced flies paused 33.7% (451/1338) compared to 34% (287/832) in DNp09/ + and 40% (223/545) in Kir2.1/ + ), suggesting that in part the effect on pausing frequency may be due to the genetic background.

**Disruption of freezing was independent of walking speed.** Given the negative relationship between speed and freezing probability mentioned above, we investigated whether silencing DNp09 neurons affected the walking speed of the flies. We found that indeed the average baseline speed was increased in silenced flies relative to controls (DNp09 > Kir2.1 = 12.81 ± 0.31 mm s$^{-1}$; DNp09/ + = 9.38 ± 0.25; Kir2.1/ + = 11.41 ± 0.26, one-way ANOVA, $F = 39.24$, $p < 0.001$. Post-hoc Tukey Hs.d. revealed a significant difference between all groups tested, $p < 0.01$. Figure 5f). This raises the possibility that the impairment seen in freezing stems from an elevation in walking speed and hence a shift in the probability of freezing behavior. To address this issue, we calculated the probability of freezing for looming stimuli occurring at different speeds. We found that despite the upward shift in speed of DNp09-silenced flies relative to controls, these flies were less likely to freeze, especially for looming stimuli occurring at lower movement speeds (Supplementary Fig. 5d). In addition, we tested DNp09-silenced and control flies in closed-loop such that looming stimuli were only presented when flies were at very low speeds (<2 mm s$^{-1}$). We found that the fraction of flies freezing was smaller relative to controls even at low movement speeds (DNp09 > Kir2.1: 30% (15/50); DNp09/ + : 88% (44/50); Kir2.1/ + : 78% (39/50). $X^2$ test, $p < 0.001$; no difference between parental controls was found, $p = 0.28$, Fig. 5g). Furthermore, DNp09-silenced flies froze less than controls (DNp09 > Kir2.1 median = 5.69, IQR = 0.59–37; DNp09/ + median = 71.73, IQR = 52.03–84.33; Kir2.1/ + median = 57.62, IQR = 36.81–82.89. Kruskal–Wallis and post-hoc Dunn tests revealed a significant difference between DNp09 > Kir2.1 and both parental controls, $p < 0.001$. No difference between parental controls was found, $p = 0.37$, Fig. 5h). Together these findings indicate that silencing DNp09 neurons directly disrupts freezing, rather than indirectly affecting freezing behavior by increasing the speed of locomotion.

**Activation of DNp09 neurons induced freezing.** If indeed DNp09 neurons are involved in the execution of freezing behavior, activating them artificially, in the absence of looming stimuli, should induce freezing. We expressed the red-shifted channelrhodopsin, CsChrimson[41], in DNp09 neurons and exposed single flies to red light using a modified version of our behavioral setup (Fig. 6a). CsChrimson requires retinal to function, thus experimental flies were raised in food containing retinal whereas control animals were raised in standard food. Flies were allowed to acclimate to the arena for 2 min. We then presented 10 trials of continuous light for 2 s, separated by 20-second intervals. We found that CsChrimson activation of DNp09 neurons was sufficient to trigger freezing in 61.5% (492/800) of trials (Fig. 6b and Supplementary Movie 4). Closer inspection of the time course of freezing induction showed that the probability of freezing increased gradually over the course of the 2-second light stimulation (Fig. 6c). The lag in freezing correlated with an initial increase in walking speed, induced by DNp09 activation (Fig. 6d). This running bout was much reduced in control flies, showing that running was mostly caused by DNp09 activity. In an unbiased behavioral representation of optogenetic activation of descending neurons it was also observed that activation of DNp09 induced running followed by pausing using different analytical methods[42]. Notably, in this study, optogenetic stimulation of DNp09 for 15 s led to sustained immobility that could last the whole stimulation period. The finding that flies first ran in response to DNp09 activation contrasts with the observation that, upon looming, flies often first paused and then ran, jumped or froze. Given that looming triggered pauses seem to be mediated by a distinct mechanism to that driving freezing, it is possible that, with looming, neurons upstream of DNp09 inhibit the initial running bout seen with artificial DNp09 activation. Finally, we observed jumps at light offset in test flies, but not control flies ($X^2$ test, $p < 0.001$, Fig. 6e). Moreover, jumps were more likely after stimulations that led to freezing than after stimulations that failed to elicit freezing ($X^2$ test, $p < 0.001$, Fig. 6e). One possible explanation for this could be that strong activation of DNp09 neurons inhibits downstream targets involved in jumping behavior, such that when DNp09 activation stops, these neurons are released from inhibition showing rebound excitation, thereby triggering the observed jumps.

**Freezing probability upon DNp09 activation depended on speed.** Given that the probability of freezing in response to looming stimuli was found to depend on the walking speed of flies at the time of threat, we asked whether the ability of DNp09 neurons in driving freezing was also modulated by walking speed. We found that the probability of freezing upon light activation of DNp09 neurons was negatively correlated with the movement speed of flies (Fig. 6f, $r^2 = 0.87$, $p = 0.007$). To confirm that DNp09-driven freezing is modulated by the movement speed of flies at the time of stimulation, we again tested flies with a closed-loop protocol, in which we controlled DNp09 activation to occur when flies were moving at low, high or very high speeds (approx. <2, >15 and >20 mm s$^{-1}$, respectively). The probability of DNp09 activation to drive freezing was highest (85%, 506/593 trials) for the flies stimulated at low movement speed and lowest (55%, 223/403 trials) for flies stimulated at high movement speed ($X^2$ test, $p < 0.001$, Fig. 6g). Together, these findings show that DNp09 neurons are a key element in the circuit mediating the speed modulation of freezing expression, and suggest that this modulation is not a result of locomotion induced changes in visual perception[43,44].

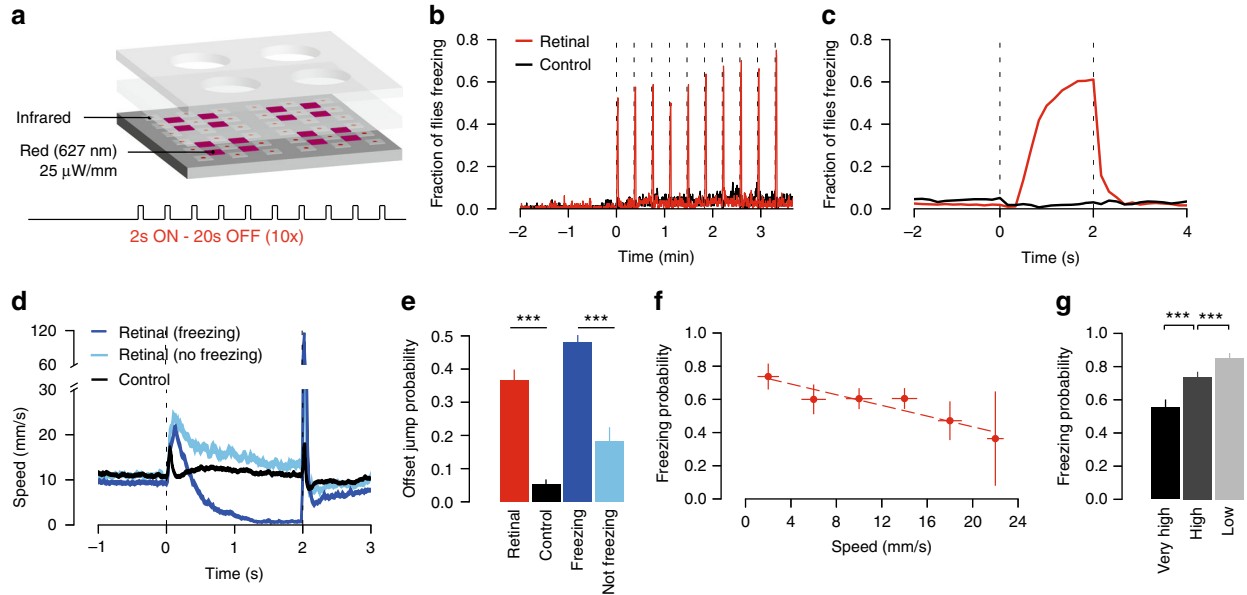

**Fig. 6** Activation of DNp09 descending neurons leads to freezing. **a** Schematic of experimental set-up for CsChrimson stimulation and stimulation protocol. Test flies supplemented with retinal (n = 80) and control flies raised in normal food (n = 72). **b** Fraction of flies freezing. Dashed lines indicate light stimulation. **c** Fraction of flies freezing aligned on light activation (retinal n = 800, control n = 720 stimulation events). **d** Stimulus-triggered speed profile (average ± s.e.m.) aligned on light presentation for stimulations that induced freezing (dark blue, n = 492), stimulations that did not (light blue, n = 308) and control (black, n = 720). **e** Probability ( + 95% CI) of jumping at light offset for control and test flies (black and red bars). Probability of jumping at light offset for stimulation events of test flies that induced freezing and events that did not (dark and light blue bars). **f** Probability of freezing to red light stimulation as a function of pre-stimulation speed ($r^2 = 0.87$, $p = 0.007$). X-error bars show s.d. for speed sampled within each interval and Y-error bars show 95% CI. **g** Probability of freezing for DNp09 > CsChrimson flies (n = 50 for each condition) tested at different speeds (number of stimulation events for very high, high and low were 403, 674 and 593, respectively). *** denotes $p < 0.001$

## Discussion

Freezing is a defensive response characterized by complete immobility, allowing prey to avoid being detected while remaining attentive to changes in the environment[36,45]. This response has been reported across vertebrates[36–38]. Although freezing has been reported previously in fruit flies[23], here we show that this behavior can be long lasting, similar to that observed in vertebrates. The pervasiveness of freezing across distant taxa strongly suggests its independent evolution and supports its adaptive value[46].

Given the prevalence of freezing behavior, it is crucially important to understand whether there are general principles that govern it. For instance, the display of freezing is plastic. Rodents tend to freeze only if there is no escape, and the presence of conspecifics decreases this behavior[4,7–9,47,48].

Furthermore, it has been previously shown that estrous cycle, feeding state and locomotor activity modulate freezing[6,11,36]. How an animal's surroundings or internal state regulate this behavior is much less clear. A hint of the conserved nature of the principles governing freezing, is the finding that in other vertebrates, such as fish, the expression of innate defensive behaviors is also plastic[10,49,50]. In this study, we extend this to invertebrate animals by demonstrating the plastic nature of freezing in flies. Flies either ran or froze in response to inescapable looming.

The choice between escaping and freezing was strongly modulated by the flies' speed at the time of threat. The effect of behavioral state on looming triggered responses could be explained by an impact of speed on motor output, sensory processing or reflects other aspects of the flies' physiology. An effect on motor output could be simply a consequence of increased difficulty in stopping when walking fast. The findings that pausing in response to looming was independent of the walking speed, and that DNp09-induced freezing was always preceded by

running, argues against an effect of movement inertia on the ability to stop. This leaves a possible influence of walking speed on visual processing[43,44,51,52] or central motor commands. It is possible that flies walking slower would have reduced visual responses to looming stimuli, leading to longer reaction time to looming, which in turn could influence the selection of defensive behaviors. However, when examining the walking speed of flies transitioning either into freezing or fleeing upon looming onset, we find similar reaction times despite the evident difference in baseline movement speed. Further, the finding that DNp09-induced freezing was modulated by movement speed of flies at the time of stimulation argues against an effect of behavioral state on sensory processing of looming. Future experiments are required to disambiguate between these scenarios.

We next explored the neuronal underpinnings of freezing behavior, contributing to the understanding of how different animals, with different bodies and brains, implement this seemingly simple behavior. We uncovered the key role of a single pair of descending neurons, DNp09, in driving freezing. DNp09 neurons innervate visual input areas in the central brain, thus being in a good position to respond to looming stimuli. The output terminals of DNp09 neurons innervate the posterior slope, which is densely innervated by other descending neurons[53], and multiple regions within the leg neuropil and tectulum, allowing the interaction with other motor outputs at different levels.

Although freezing is often seen as absence of other behaviors, or a passive state of immobility[19,45,54], evidence suggests otherwise. For example in mammals, freezing is accompanied by sustained muscle tension likely involved in postural control[55,56] and, in response to learned cues, requires sustained activity of several brain regions. In addition, a recent study identified in mice a set of descending neurons that drive stopping behavior that is distinct from those identified for freezing[18,57]. The finding

that looming-triggered freezing and pausing could be dissociated supports the idea that freezing is an active defense module pointing to the conserved nature of the distinction between freezing and stopping. Moreover, freezing may require active inhibition of alternate behaviors. An indication that active inhibition of alternate behavior happens in flies comes from our observation that DNp09-silenced flies jump more and that flies jump at the offset of DNp09 neuron activation, consistent with rebound excitation after inhibition of jump-mediating neurons. Further, the observation that jumping steeply decreased as the number of flies freezing over the course of the repeated looming increased is consistent with an inhibitory effect of freezing on jumping. Further experiments are required to definitively establish a potential active inhibition of freezing on other defensive responses. The identification of DNp09 descending neurons as central to freezing opens the path to further explore how the active state of freezing is implemented. Activation of DNp09 neurons drove both running and freezing. However, silencing DNp09 neurons left looming triggered escape responses intact suggesting that DNp09 triggered running may correspond to a different behavior. Still, it will be very interesting to unravel how a single pair of neurons drives distinct behaviors.

Finally, since the flies' speed modulates their response to looming stimuli, we examined whether the ability of DNp09 neurons to drive freezing was also modulated by the flies' speed. We found that the probability of freezing upon DNp09 stimulation was negatively correlated with the flies' movement speed. This finding demonstrated that DNp09 neurons are a key element in the circuit mediating speed dependent defensive decisions. Unraveling how speed impinges on DNp09 neurons and possibly other elements of defense circuits will be instrumental for the understanding of the organization of defensive behaviors crucial for survival.

## Methods

**Animal husbandry and fly strains**. All animals used in experiments were 4–6 days old mated female *Drosophila melanogaster*. Flies were raised at 25 °C and 70% humidity in a 12 h:12 h dark:light cycle. All behavioral experiments were performed in the 4-hour period preceding lights off and under the same conditions as rearing.

Strains and sources: Canton-S (CS) used in Figs. 1, 2, 3 and 4; CS also used to cross with parental strains in Fig. 5: *10XUAS-IVS-eGFPKir2.1* (attP2)[22] and DNp09 line[40]. DNp09 line with *20xUAS-CsChrimson.mVenus* in attp2[41]. DL flies used to cross with parental strains (same as Fig. 5) in Supplementary Fig. 6.

**Behavioral apparatus**. We recorded behavior of unrestrained flies while presenting visual stimulation (Fig. 1a). A monitor tilted at 45 degrees over the stage delivered visual stimulation. To image fly locomotion, a custom-built infrared (850 nm) LED array was placed under the stage to serve as backlight. A 2 mm white opaque acrylic sheet was placed on top of the LED array to produce homogeneous illumination. Fly behavior was recorded using a USB3 camera (PointGrey Flea3) with a 850 nm long pass filter. Behavioral arenas were custom built from opaque white and transparent acrylic sheets. Chambers were 30 mm in diameter and 4 mm in height. Single flies were aspirated into a chamber and placed on the stage. Flies were observed for 20 s to ensure that no gross motor defects were present before video acquisition was initiated.

**Visual stimulation**. Visual stimuli were presented on a 24-inch monitor (ASUS VG248QE) running at 144 Hz. All stimuli were generated in custom python scripts using PsychoPy[58]. To generate a looming effect, a black circle increased in size over a white background. The visual angle of the expanding circle was determined by the equation: $\theta(t) = 2\tan^{-1}(l/vt)$ (Eq. 1), where $l$ is half of the length of the object and $v$ the speed of the object towards the fly. Virtual object length was 1 cm and speed 25 cm s$^{-1}$ ($l/v$ value of 40 ms). Each looming presentation lasted for 500 ms. Object expanded during 450 ms until it reached maximum size of 78° where it remained for 50 ms before disappearing. Synchronous with expansion, looming stimuli produced a considerable decrease in luminance within the behavioral apparatus. We measured luminance using a digital lux meter (DX-100, INS instrumentation). When no stimulus was being presented (white screen) the luminance at the stage was 260 lux. Just before looming offset, when the disk reached its maximum size, luminance was 32 lux, representing an 88% decrease. To control for these changes, we created a stimulus where an array of approximately 5° dots was added each frame in random positions as to not create an expanding

pattern. The size and number of dots was determined empirically to generate a similar decrease in luminance as the looming stimulus (35 lux, 86.5% decrease).

**Video acquisition and tracking**. Videos were acquired using Bonsai[59] at 60 Hz and width 1104 x height 1040 resolution. Image segmentation was performed by custom software in python using OpenCV. We extracted two main features from the videos: fly position and motion activity around the fly. Positions were calculated from the centroid of an ellipse fitted to the fly by background subtraction and motion was quantified by the number of pixels active in an $100 \times 100$ pixel region of interest surrounding the fly. A pixel was considered to be active if it recorded a change higher than 10 intensity levels.

**Behavioral classifiers**. In order to automatically classify behavioral states, speed and motion tracking data were averaged into 500 ms bins and thresholds were determined by manual annotation of fly behavior (Supplementary Fig. 1). A fly was considered to be walking if its average speed exceeded 4 mm s$^{-1}$. Because a fly can exhibit low speed behaviors while not being immobile (i.e., grooming) we used pixel activity, or motion, to classify freezing bouts. A fly was considered to be freezing when average motion around the fly was lower than 50 pixels s$^{-1}$ (~5% of fly area). A minimum change of 10 intensity levels from one frame to the next was required for a single pixel to be considered active. We identified jumping events by detecting peaks in the raw, un-binned speed data. A fly was classified as having jumped if its instantaneous speed exceeded a 75 mm s$^{-1}$, a threshold identified by a discontinuity in the speed distribution. We next classified behavioral responses on a trial by trial basis (for each looming presentation, Fig. 3i). Freezing is a sustained response that can last for several looming presentations, making it difficult to assert whether a fly already freezing is responding to the looming stimulus. Still, we observed that flies freezing reliably startled upon looming, a strong indicator that flies freezing are still responding to looming. Hence, we will consider a freezing response to a given looming even if the fly initiated freezing upon a prior looming stimulus. Regarding fleeing responses, jumps are readily quantifiable, yet running bouts are difficult to assert on a single-trial basis, as an arbitrary cutoff of speed increase in such a noisy signal is unreliable. Hence, to establish the rate of fleeing responses we counted trials where flies paused and those where the fly faced away from the screen immediately after looming. A fly was classified as having paused in a walking trial if it decreased its speed below the walking threshold defined above for a consecutive period of 10 frames (160 ms, 33% of the duration of the looming, Supplementary Fig. 3b-d).

**Closed-loop looming stimulation**. Fly positions were tracked in real time and used to trigger looming stimuli using Bonsai[59]. Thresholds for looming stimuli were defined based on the displacement of the fly in 500 ms windows (Supplementary Fig. 4). Low speed loomings were triggered when displacement was smaller than 1 mm, while high speed loomings were triggered when displacement was larger than 7.5 mm. A refractory period of 15 s was imposed after each triggered stimulation and subsequent looming were shown only after the threshold was crossed again. Even though displacement was used in this experiment to trigger stimulation, speed was calculated from path length in all subsequent analysis. Behavioral arenas were built as described above, except chambers were 60 mm in diameter and 4 mm in height. Single flies were aspirated into the chamber and were allowed to explore for 2 min, then closed loop tracking was initiated and lasted for 5 min.

**Optogenetic activation**. Responses of freely moving flies to CsChrimson[41] activation were captured using the behavioral apparatus described in Fig. 6a. High-powered 627 nm LEDs were interspersed between the infrared LEDs on the backlight board. Each arena was irradiated by four red LEDs for total radiance of 0.025 mW mm$^{-2}$. Experimental flies were raised on standard fly food with 0.2 mM all trans-retinal (Sigma, R2500) and control flies were raised on standard fly food without retinal. Flies were allowed to explore for 2 min and then were stimulated with 10 repetitions of 2 s light on, 20 s light off. The rationale for this protocol was that it created a similar inter-stimulus interval as the looming stimulation. We used 2 s light on in order to elicit freezing longer than the looming-triggered pause (Fig. 3c, of ~200 ms) but not long enough to habituate neuronal activity. A stimulation event was considered successful (led to freezing) if the fly froze for more than 25% of the stimulation period (>0.5 of 2 s).

**Closed-loop optogenetic activation**. Real-time tracking and closed-loop conditions were the same as described above for the looming stimulation, except thresholds were used to trigger the red LED switch, instead of visual stimulation. Stimulations were triggered at three different displacement thresholds: smaller than 1 mm, larger than 7.5 mm and larger than 10 mm.

**Staining and imaging**. Imaging of DN morphology was performed as part of the Janelia Descending Interneuron project. The DNp09DN split-GAL4 driver line, SS1540, was crossed to *5XUAS-IVS-Syt::smGFP-HA* and *-5xUAS-IVS-myr::smGFP-FLAG* and the central nervous systems of the progeny were dissected and stained for anti-GFP according to the standard Janelia FlyLight protocol. Brains were

subsequently mounted in DPX and imaged with a confocal microscope. Detailed immunohistochemistry staining and DPX mounting protocols are available online at https://www.janelia.org/project-team/flylight/protocols. To best illustrate DNp09 morphology, off-target expression was removed from the image using Photoshop.

**Data analysis and statistics**. Data analysis was performed using custom Python scripts. All data, except those from animals excluded due to tracking errors, were analyzed. Prior to statistical testing, data were tested for normality with a Shapiro-Wilk test and the appropriate non-parametric test was chosen if data were not normally distributed. All statistical tests are specified in the results section of the text or figure captions and are two-sided. To quantify differences in reaction time in Fig. 4a, we fit the function: $f(x) = a^{(bx-c)} + d$ (Eq. 2) to the average speed trace of each trial type (running and freezing) and compared the estimates for parameter $c$ which determines the point of deceleration. We next performed a randomization test (with 5000 shuffles) to determine whether the estimates obtained for each condition were significantly different. The probability density distributions (PDF), were used to specify the probability of the random variable falling within a particular range of values, as opposed to taking on any one value. This probability is given by the integral of this variable's PDF over that range. The values shown in each graph correspond to the PDF at the bin, normalized such that the integral over the range is 1. Note that the sum of the histogram values will not be equal to 1 unless bins of unity width are chosen.

## Data availability

The data that support the findings of this study and the code used for analysis are available from the corresponding author upon reasonable request.

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

## Acknowledgements

We would like to thank the Scientific Software Platform of Champalimaud Centre for the Unknown (CCU) for developing the fly tracking software; the Scientific Hardware Platform of the CCU, for building LED array for optogenetic activation experiments; the Fly facility of the CCU and Grace Zheng from the fly facility of Janelia Research Campus; Gil Costa for the illustrations in Fig. 6a and Supplementary Figure 1a; Eugenia Chiappe and Joseph Paton for comments on the manuscript. This work was funded by the Champalimaud Foundation, the visiting scientist program of Janelia Research Campus and the ERC Starting Grant CoCO 337747. Ricardo Zacarias was supported by Fundação para a Ciência e Tecnologia SFRH/BD/51897/2012.

## Author contributions

R.Z. performed all experiments and analyzed the data. R.Z., M.L.V. and M.A.M. designed all the experiments, discussed results and wrote manuscript. S.N. and G.C. created the split-gal4 lines of descending neurons and provided the image of DNp09 neuron labeling. G.C., M.L.V. and M.A.M. supervised the unbiased DN-silencing screen performed at Janelia Research Campus; G.C. commented on the manuscript. Fly lines should be requested to G.C.

## Additional information

**Competing interests:** The authors declare no competing interests.

