## [Peer Review File · Nature Communications]

Reviewers' comments:

Reviewer #1 (Remarks to the Author):

The manuscript by Zacarias and colleagues investigates the neural basis of innate freezing behaviour in the fruit fly. Using looming stimuli as the innate threat presented in a confined arena and in combination with a clever closed-loop paradigm, the authors first find that, unexpectedly, the probability of freezing depends on the walking speed of the fly at the time of stimulus presentation. They then use gain and loss-of-function experiments to show that a pair of genetically-defined neurons - DNp09 - are important components of the innate freezing pathway. I think that the work presented in this manuscript is technically of high quality, and that the main findings reported are very interesting to the community and the wider neuroscience field, as well as timely, given a flurry of recent reports on innate defensive behaviours across a wide range of vertebrate and invertebrate species. I have however several conceptual, technical and presentation points that I would like the authors to address before publication.

Conceptual points:

1. The authors seem to be suggesting that walking speed is a readout of some sort of "state". In the abstract they refer to it as "context", in the introduction as "internal state" and later on in the discussion they relate it to "arousal". First, as the authors surely know given their past work, context and internal state can in principle refer to two very different variables, and I think that the lack of clarity muddles the interpretation of the data. If what the authors are suggesting is that flies that walk faster are more aroused, and therefore in a different internal state, then they should say so explicitly, but I would find the relationship very thin at best, as an animal can be extremely aroused and yet not moving at all (e.g.: during defensive freezing, rather ironically).
2. Is there a difference in the reaction time of stimulus detection (e.g.: startle) or behaviour onset between jumping and freezing trials? If the reaction time was longer for freezing it could support an argument around differences in arousal or awareness of the stimulus, e.g.: freezing would be initiated if the stimulus was detected too late after onset to ensure successful escape before the expected collision time.
3. The justification for focusing on freezing behaviour out of the three observed behaviours seems a bit contrived: 1) the behaviour is highly conserved - so is escape; 2) it is the main behaviour observed under the experimental conditions - the conditions are not necessarily representative of the *Drosophila* natural habitat.
4. Can the authors speculate or expand on the relative advantages between fleeing and jumping for a fly?
5. The increase in jumping after DNp09 inactivation doesn't necessarily imply inhibition of the other defensive behaviour forms - they could all have different activation thresholds and freezing just not available for selection after inactivation.
6. DNp09 activation seems to elicit running, followed by freezing, followed by jumping (which is also increased after inactivation). While I entirely follow the authors reasoning, I am left wondering if the best interpretation of these results is indeed that DNp09 neurons are critical for freezing specifically, and not part of a pathway that coordinates defensive behavioural selection.
7. On p5 the authors state that "our data suggests that flies select between ... freezing or fleeing". Note that this only applies to the current behavioural arena, where jumping escapes do not lead to a successful outcome.

Technical points:

1. The authors observe a very low percentage of jumping responses in comparison to previous

studies, and they attribute the difference to the lack of "escapability" in the arena used in this study. Why this is perfectly reasonable, it would be ideal if they could directly demonstrate this, by reproducing the higher percentage of jumping responses in a different arena, to control for other factors that might explain the differences in behaviour but have not been accounted for explicitly.

2. The control stimulus seems to elicit freezing in 12% of flies. Is this significant when compared to no stimulus presentation? If so, what is the interpretation? In a related point, can the authors provide a proper quantification in the changes in luminance between control and looming stimuli? (either by measurements or calculations).

3. Are there startle responses in jumping/fleeing trials? Or are they only seen in freezing trials?

4. The authors state that jumping events increase after DNp09 silencing, is this statistically significant? Was there any change to the frequency of fleeing events?

5. Does the running speed reached after DNp09 activation have any relationship with freezing probability?

6. Were DNp09 neurons the only neurons in the screen that had an effect on freezing? If not, why were these chosen?

7. How was the subset of neurons shown in Figure 2A chosen?

8. Do the data shown in Figure 4A all come from closed-loop experiments? (the legend is not clear). The relationship between speed and freezing probability is not linear, there seems to be a threshold at around 4-8 mm/s, can the authors comment on this profile?

9. In the DNp09 activation experiments, what was the rationale for 2s ON and 20s OFF?

10. In Figure 6F the data seemed to have been binned, shouldn't there be X-error bars? Was the correlation computed on the raw or binned data?

11. Why use displacement for the closed-loop experiments instead of track length, if the measure of interest is speed? Has speed been calculated on displacement instead of track length? I think this would be incorrect, as a fly could be moving around in circles very fast, for example.

Presentation points:

1. Throughout the presentation of the results I find it hard to get an overview of the fractions of the possible behaviours observed across all flies. The data are there, but it would make it easier for the reader if this was given explicitly, outside the data subset selection used for more specific analyses.

2. For a particular variable being quantified and compared across conditions please give summary statistics in the text, and not just the results of the statistical testing. In the current version of the text it is very hard to evaluate the magnitude of any differences between conditions.

3. I find the methods very thin, especially lacking in detail on data analysis procedures.

4. It is not ideal that often the data presentation across datasets (i.e.: between jumping, escaping and freezing) is not uniform, as it makes them hard to compare. For example, sometimes the authors report pixels changing, in others they report speed, and yet in others they report behaviour probability (e.g. 2C vs 3C vs 1D)

Reviewer #2 (Remarks to the Author):

Through an elegant repetitive looming stimulus presentation paradigm, the manuscript demonstrates that a persistent freezing (long lasting, 3 to 5 min) or fleeing response can be elicited in flies as part of their post-stimulus defensive behavioral strategy. The authors show that this post stimulus behavioral choice is predicted by the speed of flies immediately prior to looming stimulus presentation. They further characterize the neural underpinnings of this behavioral strategy and identify a key pair of P9 descending neurons.

1) Behavioral Characterization Experiments:

We appreciate the elegant design, automated scoring, and thorough analysis of single fly behaviors as presented in figures 2-4. The distinct stimulus-locked and post-stimulus behavioral profiles elicited by presentation of the same stimulus between freezing and fleeing flies is striking, but we believe that this effect could be explained by the well-documented dependence of visual perception on the walking speed in flies.

In figure 2C, the stimulus-locked startling response starts around 300ms after stimulus onset, which may suggest that the first 300ms of the looming stimulus were either not detected or attended; therefore, the stimulus may have been perceived as a more fear-inducing colliding disc rather than a looming stimulus. This is juxtaposed with the behavioral response in figure 3C, where the runner group immediately reacted to the stimulus at its onset. The difference in perception could explain the outcome of freezing vs. running behavior, rather than its being a selection by the descending system. Based on the literature, there is evidence that walk speed modulates visual responses in flies (Chiappe ME, et. al., 2010) which supports the argument here that freezers may not be detecting the visual stimulus in the first 300ms. There are many publications establishing the strong dependency of freezing/fleeing decisions on visual stimulus features (Yilmaz M and Meister M, 2013, Liden WH, et. al. 2010, Franceschi GD, et. al. 2016, Gibson WT., et. al. 2015, ...etc.). Therefore, testing different stimulus features may help to interpret the results presented here in the framework of previous publications.

The sequence of repetitive shadows has already been reported to evoke persistent defensive responses (running, freezing and jumping) within a single fly task (Gibson WT., et. al. 2015). This defensive state was shown to be active for tens of seconds (compared to 3 to 5 min for freezing state here in the manuscript), scaling with the stimulus number and frequency (Gibson WT., et. al. 2015) So, the novelty that is offered by this manuscript is that the walking speed modulates defensive responses.

2) Probing the neural basis of defensive responses:

More concerning is the validity of the claim that P9 is a key element in the circuit mediating speed dependent defensive decisions. The neural manipulation experiments, both silencing and optical activation require a serious revision.

Silencing experiments:

Figures 5E (trough region of speed profile) is not a pause response as claimed in comparison to S3A. Figures 5E and 5F provide evidence that there is an elevation in overall speed for the P9 silencing group. To address these concerns, authors provide figure S4C. However, the controls from this experiment are not consistent with each other. For instance, there are significant differences between all groups at the lowest speed. It would be useful to include wildtype as a reference to all panels in figure 5 and related supplementary figures. The other attempt to address these concerns in figure 5G suffers from a lack of sample size and high error range. The inconsistency among the controls is a general trend for every panel in figure 5. For instance, UASKir2,1/+ is exhibiting a dominant phenotype for lack of post looming stimulus-induced running (Fig 5E and S4A). We strongly suggest exchanging the background of transgenic flies to wild-type (backcross/outcross transgenes) to Canton-S since Canton-S is the reference line for figures 1-4. Or alternatively, they could use different transgenes and their combinations for the corresponding genotypes since the current alleles exhibit phenotypes that challenge the major claims. Referencing the panels to wildtypes and back-crossing may better substantiate the claims.

Optical activation experiments:

Integral to the claim that P9 mediates a choice between the defensive decisions of either fleeing or freezing is the authors' claim that P9 activation is capable of inducing running in a state-dependent manner. To support the claim that P9 induces running, authors provide figure 6D. However, the observed initial speed change is not noticeably different than red-light induced running with control flies in supplementary figure 5. Moreover, no evidence substantiates the state-dependency of P9-induced running.

Freezing was introduced as a state change that lasted 3-5 minutes after stimulus offset as shown in figure 2D. However, the activation of P9 neurons only led to a brief freezing of around 1 second during light stimulation which ceased at light offset. P9 manipulation itself is incapable of fully recapitulating the long-lasting freezing state as shown in figure 2.

Minor Concerns

1. Is fleeing a long-lasting state? In Figure 3C, how long does it take for stage 4 to return back to the baseline?
2. Figure 1C-E is not so relevant to the main claims and should be moved to supplementary figures.
3. Figure 3D is missing the control stimulus data.
4. Line 163 refers to the incorrect supplementary figure S2 (should be S3).
5. Was the head-direction accounted for during closed loop experiments since flies were tracked in real time (fig.4B, C, SFig.2, and Fig.5G)?
6. The weak off-target expression outside of P9 neuron is not shown, however; details in regards to this off-target expression might be important to report.
7. Is the freezing response "a learned response" rather than an innate as the title states? 1st looming stimulus in the sequence of 20, elicits ~25% of freezing and the likelihood of freezing climbs up to %70 at the end of the 16th stimulus. Is this learned? How many startling responses does it take to freeze?

Reviewer #3 (Remarks to the Author):

The study by Zacarias made a convincing case that there are different behavioral choices fruit flies can make to escape from looming visual stimulation, an excellent laboratory stimulation that approximates potential predators in the natural environment. Given the prevalence of escape behavior in the animal kingdom, this is an excellent example that *Drosophila* is a lovely genetic system that can provide important insights into potentially evolutionary conserved mechanisms. The authors first showed that flies either freeze or flee to escape from the looming stimulus. They then showed that a fly's initial walking speed is well correlated with its behavioral choice to flee or to freeze. And silencing a pair of descending neurons eliminates the freezing behavior but leaves the running behavior intact. Overall, the manuscript is well written and experimental results provide a reasonable support for the thesis of the study. I only have some minor concerns.

1. There is a need to present the three looming-induced behaviors in reference to the common behavioral state of a fly. This is important for understanding how context may bias behavioral choice. Results from Figure 4 show that the initial walking speed is correlated with the freezing probability – slow walking flies are more likely to exhibit freezing behavior. These results and those in Figure 3 suggest that fast walking flies would likely to run away from the looming stimulation. But it is not clear whether jumping behavior, another way to flee the looming stimulus, is biased by the initial walking speed too. Moreover, it is not at clear whether pause precedes jumping and freezing behavior like running behavior. These are important characterization that the authors have the data and it may help to build a model about the decision making process of flies to escape from the being detected by potential predators.

2. Based on the observation that the probability of freezing behavior increases with repeated looming stimulations, the authors argue that there is no habituation in the process. However, there is insufficient data for this statement. There are three different escape behavior characterized here – running, jumping, and freezing. Whether there is habituation or not, the authors need to examine the total probability of all three behaviors.

Dear editor and reviewers,

Please find enclosed our point-to-point reply to the reviewers' comments, which we believe have greatly contributed to improve our manuscript.

Kind regards,

The authors

Reviewer #1 (Remarks to the Author):

The manuscript by Zacarias and colleagues investigates the neural basis of innate freezing behaviour in the fruit fly. Using looming stimuli as the innate threat presented in an confined arena and in combination with a clever closed-loop paradigm, the authors first find that, unexpectedly, the probability of freezing depends on the walking speed of the fly at the time of stimulus presentation. They then use gain and loss-of-function experiments to show that a pair of genetically-defined neurons - DNp09 - are important components of the innate freezing pathway. I think that the work presented in this manuscript is technically of high quality, and that the main findings reported are very interesting to the community and the wider neuroscience field, as well as timely, given a flurry of recent reports on innate defensive behaviours across a wide range of vertebrate and invertebrate species. I have however several conceptual, technical and presentation points what I would like the authors to address before publication.

Conceptual points:

1. The authors seem to be suggesting that walking speed is a readout of some sort of "state". In the abstract they refer to it as "context", in the introduction as "internal state" and later on in the discussion they relate it to "arousal". First, as the authors surely know given their past work, context and internal state can in principle refer to two very different variables, and I think that the lack of clarity muddles the interpretation of the data. If what the authors are suggesting is that flies that walk faster are more aroused, and therefore in a different internal state, then they should say so explicitly, but I would find the relationship very thin at best, as an animal can be extremely aroused and yet not moving at all (e.g.: during defensive freezing, rather ironically).

We agree with the reviewer that context may be interpreted by many researchers, especially in the fear conditioning field, as the surrounding environment. Therefore, we have changed the manuscript to refer to this variable as 'behavioral state'. We prefer to keep it as agnostic as possible in regards to what exactly this state constitutes as our experiments do not address this issue directly. It can be an arousal state, however as the reviewer points out locomotion is probably not the best proxy for arousal, although commonly used in the *Drosophila* literature. Better measures of arousal in flies are required to further explore its relationship with defensive behaviors.

We have modified the abstract (line 16), to read 'The probability of freezing versus fleeing was modulated by the fly's walking speed at the time of threat, demonstrating that freeze/flee decisions depended on behavioral state.'

We have substantially changed the third paragraph of the discussion to reflect this clarification point (lines 360-3).

2. Is there a difference in the reaction time of stimulus detection (e.g.: startle) or behaviour onset between jumping and freezing trials? If the reaction time was longer for freezing it could support an argument around differences in arousal or awareness of the stimulus, e.g.: freezing would be initiated if the stimulus was detected too late after onset to ensure successful escape before the expected collision time.

We thank the reviewer for suggesting stimulus detection time as possible explanation for the selection between fleeing and freezing responses. We have examined reaction time to looming in trials where flies froze or fled in response to looming, by plotting the speed of flies before, during, and after the looming stimulus. It is possible to estimate the moment of looming detection in freezing trials as there is a sharp deceleration during the looming stimulus. For escape trials the onset of the pause can be used to determine time of detection. We found no obvious difference in the onset of deceleration in both freezing and escape trials. This result is now reported as Figure 4A and discussed in the first paragraph of the 'Freezing/fleeing decisions were modulated by the walking speed of the flies at the time of threat' section (lines 184-188).

Furthermore, the finding that optogenetic activation of DNp09 drives freezing in a state dependent manner argues against an effect on looming detection.

3. The justification for focusing on freezing behaviour out of the three observed behaviours seems a bit contrived: 1) the behaviour is highly conserved - so is escape; 2) it is the main behaviour observed under the experimental conditions - the conditions are not necessarily representative of the *Drosophila* natural habitat.

We have changed the beginning of the section 'Activity of DNp09 descending neurons is required for freezing but not fleeing in response to looming' to read 'Next, we searched for the neural mechanisms underlying the defensive behaviors observed. We focused on freezing, as it corresponds to the dominant behavior adopted by flies in our experimental conditions, it is readily quantifiable and though it is conserved across the animal kingdom (1–3) very little is known regarding the neural mechanisms of freezing in insects.' to better justify focusing on freezing behavior.

4. Can the authors speculate or expand on the relative advantages between fleeing and jumping for a fly?

We believe that the two fleeing responses, running and jumping can be adaptive in different environments. Jumps are likely take off attempts allowing the fly to fly away from the threat, while escape runs might constitute an attempt to reach refuge. Hence, flying may be advantageous in an open environment while escape runs might be better in cluttered ones. Since our study has a mechanistic focus, both at the behavioral and neuronal level, we prefer not to include this speculation on adaptive value of the two fleeing behaviors when so little is known about the natural behavior of flies.

5. The increase in jumping after DNp09 inactivation doesn't necessarily imply inhibition of the other defensive behaviour forms - they could all have different activation thresholds and freezing just not available for selection after inactivation.

Although the reviewer is right regarding the interpretation of the effect of DNp09 silencing on jumping frequency, our speculation regarding a possible role of these neurons in directly inhibiting jumps comes from the combined results of the silencing and the optogenetic activation experiments. As we say in the discussion: 'An indication that active inhibition of alternate behavior happens in flies comes from our observation that DNp09 silenced flies jump

more and that flies jump at the offset of DNp09 neuron activation, presumably resulting from rebound excitation after inhibition of jump. Consistent with an inhibitory effect of freezing on jumping is the observation that jumping steeply decreased with increase number of flies freezing over the course of the repeated looming stimuli.'

To emphasize the speculative nature of our statement regarding a possible role of DNp09 in inhibiting jumps, we added in the discussion the following sentence: 'Further experiments are required to definitively establish an active inhibition of other defensive responses.' (lines 390-1).

6. DNp09 activation seems to elicit running, followed by freezing, followed by jumping (which is also increased after inactivation). While I entirely follow the authors reasoning, I am left wondering if the best interpretation of these results is indeed that DNp09 neurons are critical for freezing specifically, and not part of a pathway that coordinates defensive behavioural selection.

We agree with the reviewer and thus state in the conclusion paragraph that '(...) This finding demonstrated that DNp09 neurons are a key element in the circuit mediating speed dependent defensive decisions'. However, given the inactivation results, we believe there is an asymmetry in the involvement of this neuron in the defensive responses observed, there could be more redundancy in the circuitry that drives running compared to that driving freezing.

7. On p5 the authors state that "our data suggests that flies select between ... freezing or fleeing". Note that this only applies to the current behavioural arena, where jumping escapes do not lead to a successful outcome.

We have changed that sentence to read: 'Our data suggest that in our experimental conditions flies select between two distinct behavioral strategies, freezing or fleeing.' (lines 183-184).

Technical points:

1. The authors observe a very low percentage of jumping responses in comparison to previous studies, and they attribute the difference to the lack of "escapability" in the arena used in this study. Why this is perfectly reasonable, it would be ideal if they could directly demonstrate this, by reproducing the higher percentage of jumping responses in a different arena, to control for other factors that might explain the differences in behaviour but have not been accounted for explicitly.

To address the reviewer's concern we performed a new experiment, where flies were exposed to the same looming stimulus in the same set-up but could fly away in response to the stimuli. To this end, we allowed flies to climb to the top of the arena and then presented a looming stimulus. In this condition, where looming was escapable, 90% (36/40) of flies jumped in response to looming. This result was added to the manuscript in the section 'Flies jumped rarely in response to repeated inescapable looming' of the results. (lines 88-91).

2. The control stimulus seems to elicit freezing in 12% of flies. Is this significant when compared to no stimulus presentation? If so, what is the interpretation? In a related point, can the authors provide a proper quantification in the changes in luminance between control and looming stimuli? (either by measurements or calculations).

We have added the experimental condition where no stimulus was presented (we ran these experiments in parallel with the others but had previously not included them in manuscript). The results of this experiment are now plotted together with looming and control stimulus conditions in supplementary figure 2.

We now report the luminance in the arena with white screen, final looming size and final dot size, in the methods section (lines 587-593). The looming stimulus generates an 88% decrease in luminance and the neutral stimulus generates an 86.5% decrease in luminance.

3. Are there startle response in jumping/fleeing trials? Or are they only seen in freezing trials?

The way we define the startle response, as a sudden increase in pixel change during looming, was only applicable to flies that were freezing. This startle response, which we believe might be an aborted jump, may be present in other trials but it is difficult to assert with certainty.

4. The authors state that jumping events increase after DNp09 increase after silencing, is this statistically significant? Was there any change to the frequency of fleeing events?

The increase in jumping frequency seen in DNp09 silenced flies is statistically significant (Kruskal-Wallis test, $p < 0.001$), and is reported in the current supplementary figure 5C. We have added an analysis on the fraction of flies freezing, jumping or running throughout the 20 looming presentations. We found that, in control flies, the rate of running decreases as the fraction of flies freezing increases. This is not seen in DNp09 silenced flies, which show a constant rate of running across the entire stimulation period. These plots are presented in supplementary figure 5E.

5. Does the running speed reached after DNp09 activation have any relationship with freezing probability?

The running speed reached upon DNp09 activation is correlated with the probability of flies freezing. However, the peak running speed during activation is also correlated with the speed prior to activation, therefore the meaning of this correlation is difficult to interpret. Since looming-triggered freezing is modulated by the walking speed prior to looming, despite the absence of a running bout, we believe the correlation between peak speed during DNp09 activation and freezing results from the relationship between speed prior to activation and freezing.

6. Were DNp09 neurons the only neurons in the screen that had an effect on freezing? If not, why were these chosen

We found a few lines that affected freezing. These results were not confirmed except for the DNp09 line, which we chose, because it was a clean line with interesting anatomy.

7. How was the subset of neurons shown in Figure 2A chosen?

Figure 2A shows a random subset of flies. Please find below other rasters with different subsets of flies showing very similar results.

8. Do the data shown in Figure 4A all come from closed-loop experiments? (the legend is not clear). The relationship between speed and freezing probability is not linear, there seems to be a threshold at around 4-8 mm/s, can the authors comment on this profile?

To clarify where the data comes from we have now added a sentence to the figure legend that reads: ‘panels A and B show data, already used in Figures 1-3, from open-loop experiments, whereas panels C-E show data from close-loop experiments’. Indeed the relationship between speed and freezing is not linear. There is a big effect of behavioral state, not walking/walking, but also one of walking speed that is however less pronounced.

9. In the DNp09 activation experiments, what was the rationale for 2s ON and 20s OFF?

We used a inter activation interval of 20s for the optogenetic experiments that was similar to inter stimulus interval of looming stimulation. 2s ON was a duration that would be sufficient to see immobility but not too long such that it could lead to inactivation of DNp09 neurons. We did not play around much with stimulation parameters, as our goal was just to see if these neurons could drive immobility, and not necessarily to recapitulate fully its natural function, which is more often than not fails. However, Cande et al (BioRxiv 10.1101/230128) have stimulated these neurons for 15 seconds and seen sustained immobility.

10. In Figure 6F the data seemed to have been binned, shouldn't there be X-error bars? Was the correlation computed on the raw or binned data?

We have now added error bars to this figure.

11. Why use displacement for the closed-loop experiments instead of track length, if the measure of interest is speed? Has speed been calculated on displacement instead of track length? I think this would be incorrect, as a fly could be moving around in circles very fast, for example.

This originally resulted from an error in coding the experiment. However, because there was a strong correlation between track length and displacement with this method we achieved the desired manipulation, i.e. one group of flies was presented with looms at low movement speeds and another at high speeds. We have now added a panel to supplementary figure 4 showing the full distribution of pre-loom speeds for both groups of flies. Very little overlap is found between the two. When analyzing movement speed of flies we calculated the actual speed based on trajectory length rather than displacement.

Presentation points:

1. Throughout the presentation of the results I find it hard to get an overview of the fractions of the possible behaviours observed across all flies. The data are there, but it would make it easier for the reader if this was given explicitly, outside the data subset selection used for more specific analyses.

We have now added new plots that provide an overview of the rate of the different response and non-responders over the course of the 20 looming presentations (Figures 3I and Supplementary Figure 5E).

2. For a particular variable being quantified and compared across conditions please give summary statistics in the text, and not just the results of the statistical testing. In the current version of the text it is very hard to evaluate the magnitude of any differences between conditions.

Done.

3. I find the methods very thin, especially lacking in detail on data analysis procedures.

We have changed the methods section extensively to provide more detail.

4. It is not ideal that often the data presentation across datasets (i.e.: between jumping, escaping and freezing) is not uniform, as it makes them hard to compare. For example, sometimes the authors report pixels changing, in others they report speed, and yet in others they report behaviour probability (e.g. 2C vs 3C vs 1D)

We have used different metrics for the different behaviors as appropriate. For example, if a fly is freezing, its centroid (which we use to measure the fly's speed) does not move, thus rendering speed not very informative as a measure. Because startle responses are usually small movements with very little centroid displacement, the number of pixels changing around the fly provides us with a better quantification of motion in place. We have now added Figure 3I in which we plot the probabilities of each behaviors within the same axes. This representation may help in making comparisons among the different behaviors described.

Reviewer #2 (Remarks to the Author):

Through an elegant repetitive looming stimulus presentation paradigm, the manuscript demonstrates that a persistent freezing (long lasting, 3 to 5 min) or fleeing response can be elicited in flies as part of their post-stimulus defensive behavioral strategy. The authors show that this post stimulus behavioral choice is predicted by the speed of flies immediately prior to

looming stimulus presentation. They further characterize the neural underpinnings of this behavioral strategy and identify a key pair of P9 descending neurons.

1) Behavioral Characterization Experiments:

We appreciate the elegant design, automated scoring, and thorough analysis of single fly behaviors as presented in figures 2-4. The distinct stimulus-locked and post-stimulus behavioral profiles elicited by presentation of the same stimulus between freezing and fleeing flies is striking, but we believe that this effect could be explained by the well-documented dependence of visual perception on the walking speed in flies.

Indeed, changes in visual perception induced by walking speed, could affect stimulus detection time. This could explain the selection between fleeing and freezing responses. Please see reply to reviewer 1 comment #2. We have added a new figure panel (Figure 4A) and a brief discussion in the first paragraph of the 'Freezing/fleeing decisions were modulated by the walking speed of the flies at the time of threat' section (lines 183-188).

In figure 2C, the stimulus-locked startle response starts around 300ms after stimulus onset, which may suggest that the first 300ms of the looming stimulus were either not detected or attended; therefore, the stimulus may have been perceived as a more fear-inducing colliding disc rather than a looming stimulus.

Although startle responses have long latencies, stopping upon looming takes place almost immediately upon looming onset. Hence startle responses are unlikely to reflect the fly's latency to detect the stimulus. Because startle responses were quantified in flies already freezing, it is not possible to determine latency to looming detection in those trials.

This is juxtaposed with the behavioral response in figure 3C, where the runner group immediately reacted to the stimulus at its onset. The difference in perception could explain the outcome of freezing vs. running behavior, rather than its being a selection by the descending system. Based on the literature, there is evidence that walk speed modulates visual responses in flies (Chiappe ME, et. Al., 2010) which supports the argument here that freezers may not be detecting the visual stimulus in the first 300ms. There are many publications establishing the strong dependency of freezing/fleeing decisions on visual stimulus features (Yilmaz M and Meister M, 2013, Liden WH, et.

al. 2010, Franceschi GD, et. Al. 2016, Gibson WT., et. Al. 2015, ...etc.). Therefore, testing different stimulus features may help to interpret the results presented here in the framework of previous publications.

Please see reponse to the first paragraph of your comments and reply to reviewer 1 comment #2.

The sequence of repetitive shadows has already been reported to evoke persistent defensive responses (running, freezing and jumping) within a single fly task (Gibson WT., et. al. 2015). This defensive state was shown to be active for tens of seconds (compared to 3 to 5 min for freezing state here in the manuscript), scaling with the stimulus number and frequency (Gibson WT., et. al. 2015) So, the novelty that is offered by this manuscript is that the walking speed modulates defensive responses.

Indeed Gibson et al report that flies freeze in response to a translational shadow. The freezing we observe is more frequent and more sustained. These differences may originate from the different stimuli used in our study and that of Gibson et al. Interestingly, Franceschi et al.

compared freezing and escape responses to looming and translational stimuli in mice and found that these stimuli triggered a different rate of freezing and escaping. Still, we do agree with the reviewer that one strong point of our systematic analysis of defensive behaviors in our paradigm is the modulation of freezing by movement speed of flies.

2) Probing the neural basis of defensive responses:

More concerning is the validity of the claim that P9 is a key element in the circuit mediating speed dependent defensive decisions. The neural manipulation experiments, both silencing and optical activation require a serious revision.

Silencing experiments:

Figures 5E (trough region of speed profile) is not a pause response as claimed in comparison to S3A.

Figure 3C and figure 5E represented the velocity around looming stimuli for all trials where flies were not freezing. This included both trials where flies paused and trials where they didn't. The fact that on half of the trials, flies paused, made the trough region of the speed profile conspicuous. This is how we became aware of the pausing behavior. In figure S3A, we were focusing on showing that the decrease in speed was really resulting from pausing and therefore separated trails with and without pauses. Hence the comparison we meant to highlight was between Figures 3C and 5E, and not figures 5E and S3A.

We now realize how this can be confusing and we thank the reviewer for pointing this out. The reviewer's comment and those of reviewer one on the frequency of pausing has sparked a drive to re-examine the characterization of pausing for each looming event. In supplementary Figure 3 D-E we show how a pause threshold was defined, which was then used to precisely quantify the rate of pausing and classify pausing and non-pausing trials. With this new analysis it is clear that DNp09 silenced flies show clear pausing responses albeit at a decreased frequency.

Figures 5E and 5F provide evidence that there is an elevation in overall speed for the P9 silencing group. To address these concerns, authors provide figure S4C. However, the controls from this experiment are not consistent with each other. For instance, there are significant differences between all groups at the lowest speed. It would be useful to include wildtype as a reference to all panels in figure 5 and related supplementary figures.

We have redone all experiments with new controls where the parental lines were crossed to wild type CS flies. The parental controls in this case show a very consistent behavior. For this reason and because all other experiments were performed using CS flies, we chose to include the experiment with CS flies in the main figure and moved the experiments with DL flies to supplementary section (Figure S6).

The other attempt to address these concerns in figure 5G suffers from a lack of sample size and high error range.

The high error range reflects the bimodality of the behavior. For consistency with figures 2B and 4D we have added a plot of the fraction of flies freezing over the stimulation period, which clearly shows an effect of silencing DNp09 even when looming was presented at low movement speeds.

The inconsistency among the controls is a general trend for every panel in figure 5. For instance, UASKir2,1/+ is exhibiting a dominant phenotype for lack of post looming stimulus-

induced running (Fig 5E and S4A). We strongly suggest exchanging the background of transgenic flies to wild-type (backcross/outcross transgenes) to Canton-S since Canton-S is the reference line for figures 1-4. Or alternatively, they could use different transgenes and their combinations for the corresponding genotypes since the current alleles exhibit phenotypes that challenge the major claims. Referencing the panels to wildtypes and back-crossing may better substantiate the claims.

We have followed the reviewer's suggestion as mentioned two points above.

Optical activation experiments:

Integral to the claim that P9 mediates a choice between the defensive decisions of either fleeing or freezing is the authors' claim that P9 activation is capable of inducing running in a state-dependent manner. To support the claim that P9 induces running, authors provide figure 6D. However, the observed initial speed change is not noticeably different than red-light induced running with control flies in supplementary figure 5. Moreover, no evidence substantiates the state-dependency of P9-induced running.

We see how control and test condition induced running may seem comparable when presented in separate plots. However, when presented in the same plot it becomes clear that peak and duration of running are very different. We have included the control condition in panel 6D to make our point more clear.

It was not our intention to make the claim that DNp09-induced running is state dependent. We have double-checked the manuscript to make sure that no claims of state-dependency of running are made. We focused this kind of analysis on freezing behavior where we show a clear correlation between freezing probability and the flies walking speed prior to stimulation.

Our view is that running induced by DNp09 activation is different from running in response to looming because running in response to looming remains intact when DNp09 is silenced. This indicates that there are multiple pathways to initiate running. As running initiation by DNp09 activation is not clearly involved in looming response, its characterization is beyond the scope of this manuscript.

Freezing was introduced as a state change that lasted 3-5 minutes after stimulus offset as shown in figure 2D. However, the activation of P9 neurons only led to a brief freezing of around 1 second during light stimulation which ceased at light offset. P9 manipulation itself is incapable of fully recapitulating the long-lasting freezing state as shown in figure 2.

Although we agree with the reviewer that our optogenetic experiments do not fully recapitulate looming triggered behavior, this was not our expectation. We think that looming activates a large number of neurons that together orchestrate looming triggered responses, whether freezing or fleeing. Here, we do report freezing bouts that can last 3 to 5 minutes, but by no means this is its definition. Freezing was defined as immobility that lasted for at least 500ms. However, when triggered it normally lasted far more than 500ms. We have chosen a brief stimulation protocol because we feared longer activation periods could lead to the inactivation of DNp09 neurons rather than their sustained activation. However, Cande et al. have stimulated DNp09 neurons for 15 seconds and found brief running bouts followed by sustained freezing up to the duration of stimulation(4). We have added this citation, which we feel addresses in part the reviewer's concern. We have also made sure that in the manuscript we do not claim that activating DNp09 optogenetically recapitulates looming-triggered long lasting freezing.

Minor Concerns

1. Is fleeing a long-lasting state? In Figure 3C, how long does it take for stage 4 to return back to the baseline?

We believe that in our experiments the effect of looming on locomotion has two components, one comprised of a brief fast escape response (away from the loom), and a more sustained state change reflected in a sustained increase in walking speed throughout the stimulation period. Below we plot the walking speed of flies exposed to 5 looming stimuli, such that the decay in walking speed after looming can be better assessed. We found that walking speed stays elevated (box plot on the right shows walking speed 1 minute before looming stimulation, white box, and the last minute of the test session, grey box). Still, as this is out of the scope of our manuscript we decided not to include it.

2. Figure 1C-E is not so relevant to the main claims and should be moved to supplementary figures.

Jumping in response to looming stimuli, with the exception of Gibson et al., is the most commonly measured response. We decided to keep the figure showing this response so the reader has an immediate sense of how our responses compare to previous work.

3. Figure 3D is missing the control stimulus data.

We thank the reviewer for pointing this out. We have added the control data to the figure panel (current Figure 3F).

4. Line 163 refers to the incorrect supplementary figure S2 (should be S3).

Thank you for calling our attention to this mistake. We have changed and double-checked all references to figures in the paper.

5. Was the head-direction accounted for during closed loop experiments since flies were tracked in real time (fig.4B, C, SFig.2, and Fig.5G)?

We have not taken head position into account as we found that movement speed was highly predictive of freezing probability and this was the feature we wanted to test further.

6. The weak off-target expression outside of P9 neuron is not shown, however; details in regards to this off-target expression might be important to report.

The images of GFP expression of the DNp09 line is now available online. We have added the link so that all readers can verify it directly. (line 234).

7. Is the freezing response “a learned response” rather than an innate as the title states? 1st looming stimulus in the sequence of 20, elicits ~25% of freezing and the likelihood of freezing climbs up to %70 at the end of the 16th stimulus. Is this learned? How many startling responses does it take to freeze?

We thank the reviewer for pointing out the emphasis implied in the title on the innate nature of freezing. We believe that freezing can be innately triggered by a looming stimulus, hence require no prior exposure to it. The amount and probability of freezing can be modulated by several factors including the external environment, which may be learned. This learning could explain the increase in freezing over the course of stimulation. Alternatively, as flies are exposed to looming their internal state changes, which in turn can lead to increased freezing, not necessarily involving learning. As we do not wish take a strong position in this regard we have removed the word innate from the title.

Reviewer #3 (Remarks to the Author):

The study by Zacarias made a convincing case that there are different behavioral choices fruit flies can make to escape from looming visual stimulation, an excellent laboratory stimulation that approximates potential predators in the natural environment. Given the prevalence of escape behavior in the animal kingdom, this is an excellent example that *Drosophila* is a lovely genetic system that can provide important insights into potentially evolutionary conserved mechanisms. The authors first showed that flies either freeze or flee to escape from the looming stimulus. They then showed that a fly’s initial walking speed is well correlated with its behavioral choice to flee or to freeze. And silencing a pair of descending neurons eliminates the freezing behavior but leaves the running behavior intact. Overall, the manuscript is well written and experimental results provide a reasonable support for the thesis of the study. I only have some minor concerns.

1. There is a need to present the three looming-induced behaviors in reference to the common behavioral state of a fly. This is important for understanding how context may bias behavioral choice. Results from Figure 4 show that the initial walking speed is correlated with the freezing probability – slow walking flies are more likely to exhibit freezing behavior. These results and those in Figure 3 suggest that fast walking flies would likely to run away from the looming stimulation. But it is not clear whether jumping behavior, another way to flee the looming stimulus, is biased by the initial walking speed too.

We have analyzed the relationship between movement speed and jumping. Interestingly, we found a U-shaped curve (see below). Because jumps are rare in our experiments, occurring in only 6,4% of the trials, and given the already lengthy and detailed description of the dominant behaviors in our experiments, we decided to not include this information in the manuscript.

Moreover, it is not at clear whether pause precedes jumping and freezing behavior like running behavior. These are important characterization that the authors have the data and it may help to build a model about the decision making process of flies to escape from the being detected by potential predators.

A pause can only be detected for flies walking before the looming stimulus. This means any analysis on pausing we may make describes only part of the trials. For instance only 37% (142/384) of the jumps occurred in trials where the flies were walking before looming stimulus. Of these 64.8% (92/142) were preceded by pausing. For analysis of pausing before freezing, even for trials (see graph below) where we can observe a drop in speed equivalent to that of trials where there is a pause, we hesitate to call it pausing as we cannot be sure if it is not a freezing bout interrupted by a startle. Although we do agree with the reviewer that it is very interesting, until we have a better handle on pausing behavior, we prefer to not over elaborate on the potential hierarchical relationship between pausing and the other three responses.

2. Based on the observation that the probability of freezing behavior increases with repeated looming stimulations, the authors argue that there is no habituation in the process. However, there is insufficient data for this statement. There are three different escape behavior characterized here – running, jumping, and freezing. Whether there is habituation or not, the authors need to examine the total probability of all three behaviors.

We thank the reviewer for bringing up this point. We have added a new analysis to address this concern (also by reviewer 1) that we believe has strengthened the manuscript, by providing a global view of the occurrence of the different defensive responses we observed over the course of looming stimulation (see Figures 3I and supplementary 5E). We found that the decrease in the fraction of flies escaping is tightly coupled with the increase in the fraction of flies freezing. Importantly, we see a very faint increase in non-responders, arguing against a habituation process.

REVIEWERS' COMMENTS:

Reviewer #1 (Remarks to the Author):

The authors have significantly revised the manuscript to address my comments. The text has been revised to clarify issues with the previous version, new analysis supporting the main conclusions has been added, the methods have been extended and new experimental data have been incorporated to substantiate the arguments. I find the paper now suitable for publication, my only remaining comment is that a statistical test should be provided for the new analysis shown in Figure 4A, on the lack of differences in reaction time.

Reviewer #2 (Remarks to the Author):

Please check my replies (in magenta) in the attached file (Dear Editor and Authors.pdf).

In addition, I am attaching another pdf with some comments on the summary and introduction sections of your MS, but these suggestions I have are broadly applicable to the rest of the MS. I would suggest a serious line-by-line editing of the MS.

Besides, Figure 5's title is wrong, Figure 6 is missing legends.

I would suggest removing Supplementary Figure 6 for consistency across the study.

Reviewer #3 (Remarks to the Author):

In the revised manuscript, the authors addressed all my previous concerns. I am therefore supporting its publication.

Reviewer #1 (Remarks to the Author):

The authors have significantly revised the manuscript to address my comments. The text has been revised to clarify issues with the previous version, new analysis supporting the main conclusions has been added, the methods have been extended and new experimental data have been incorporated to substantiate the arguments. I find the paper now suitable for publication, my only remaining comment is that a statistical test should be provided for the new analysis shown in Figure 4A, on the lack of differences in reaction time.

We thank the reviewer for this request, which we believe strengthens our claim. We have now included a shuffle test to test the difference in the point speed deceleration. In detail: To quantify differences in reaction time in Figure 4a, we fit the function: $f(x) = a e^{(bx-c)} + d$ (2) to the average speed trace of each trial type (running and freezing) and compared the estimates for parameter c which determines the point of deceleration. We next performed a randomization test (with 5000 shuffles) to determine whether the estimates obtained for each condition were significantly different. (Lines 621-5 of the methods section). We found no difference between the reaction times, defined by the deceleration point, between running and freezing trials. (Lines 175-7 of results section)

Reviewer #2 (Remarks to the Author):

Please check my replies (in magenta) in the attached file (Dear Editor and Authors.pdf).

In addition, I am attaching another pdf with some comments on the summary and introduction sections of your MS, but these suggestions I have are broadly applicable to the rest of the MS. I would suggest a serious line-by-line editing of the MS.

We have done an extensive line-by line editing of the MS.

Besides, Figure 5's title is wrong, Figure 6 is missing legends.

This has been corrected.

I would suggest removing Supplementary Figure 6 for consistency across the study.

We believe Supplementary Figure 6 strengthens our results since it reveals similar results in a different background, therefore we decided to keep it.

Reviewer #2 comment:

I appreciate the new additions pertaining to the concerns. Authors argue that the concurrent, stimulus-locked pausing response in groups with different baseline speeds (approx.. 7 vs 11 mm/s) suggests that walking speed is not exerting an effect on detection of the threat. However, as the freezing

probabilities in figure 4B indicate, at the two baseline speeds in figure 4A, there is no significant difference. Therefore, any difference in visual perception could not be captured by looking at the baseline speeds in figure 4A. To substantiate the authors' claims (e.g. lines 360-364, etc.), a comparison of pausing response to stimulus for two groups with baseline speeds that significantly differ in freezing probability would be beneficial. Since disambiguating the effect of walking speed on visual perception cannot be ruled out, I invite authors to make claims sparingly in this regard.'

We agree with the reviewer that taking our results from the behavioral analysis on reaction times to looming, one could still hypothesize that flies walking slower could have a higher looming detection threshold. However, as argued in the discussion we see a modulation of optogenetically-induced freezing by the flies walking speed at the time of stimulation (in these experiments freezing was not evoked by a visual stimulus). Our results point towards an effect of walking speed that cannot be fully explained by an impact on visual processing. We acknowledge that our study does not fully address the mechanism by which walking speed affects freezing by stating that further experiments are required to achieve this goal. (Lines 333-4 of the discussion section)